

# A propensity score-matched retrospective study on sacubitril/valsartan use and 6-month readmission rates in patients with acute heart failure

Lingdi Zhang, Qinglian Chen and Guofei Ren

Pharmacy Department, Affiliated Xiaoshan Hospital, Hangzhou Normal University, Hangzhou, China

## ABSTRACT

**Background:** Sacubitril/valsartan is a compound preparation commonly used for treating chronic heart failure with reduced ejection fraction. In clinical practice, it is also administered to patients with newly diagnosed acute heart failure. However, limited research exists on its benefits during hospitalization in this patient population.

**Aim:** To investigate the impact of in-hospital use of sacubitril/valsartan on the 6-month readmission rate in patients with newly diagnosed acute heart failure.

**Method:** A retrospective study was conducted involving 176 patients admitted for the first time with acute heart failure at Xiaoshan Hospital Affiliated to Hangzhou Normal University. Patients were divided into two groups based on whether they received sacubitril/valsartan during hospitalization: a treatment group and a control group. Clinical data were collected, and propensity score matching (PSM) was used to balance baseline characteristics. Logistic regression analysis was performed to evaluate the effect of sacubitril/valsartan on 6-month readmission.

**Results:** After PSM, 53 matched pairs were obtained, with balanced covariates between the two groups. Before matching, the readmission rate was 10.47% in the treatment group and 25.56% in the control group. Multivariate logistic regression indicated that sacubitril/valsartan was associated with a reduced risk of readmission (OR = 0.339, 95% CI [0.116–0.994], $P$ = 0.049). However, after matching, the readmission rate was 13.21% in the treatment group and 20.75% in the control group. Logistic regression analysis post-matching showed no statistically significant reduction in readmission risk (OR = 0.728, 95% CI [0.212–2.496], $P$ = 0.252).

**Conclusions:** In-hospital use of sacubitril/valsartan does not significantly reduce the risk of 6-month readmission in patients with newly diagnosed acute heart failure.

## INTRODUCTION

Sacubitril/valsartan, also known as sacubitril valsartan sodium tablets, contains two components: sacubitril and valsartan. Sacubitril is an enkephalin inhibitor that can inhibit the degradation of natriuretic peptide to increase the natriuretic peptide in the body, thus

Corresponding author
Guofei Ren, qianshuilian@126.com

promoting natriuretic diuresis. Valsartan is an angiotensin II receptor antagonist that lowers blood pressure, inhibits myocardial remodeling, and improves myocardial function. Sacubitril/valsartan is approved for chronic heart failure with reduced ejection fraction, reducing the risk of cardiovascular death and hospitalization due to heart failure. However, in clinical practice sacubitril/valsartan is also used in newly diagnosed acute heart failure (AHF) in addition to chronic heart failure. There is little research on the benefit of sacubitril/valsartan used in newly diagnosed AHF during hospitalization (*Senni et al., 2020*). We conducted a retrospective observational study to determine whether AHF could benefit from sacubitril/valsartan. The impact of using sacubitril/valsartan during hospitalization on the readmission rate in patients with AHF was evaluated. As an observational study bias and other issues are inevitable, which can affect the accuracy of the results. Therefore, the propensity score matching (PSM) method was involved in the study to minimize bias.

PSM method is a statistical method first proposed in 1983, mainly used for subgroup analysis of observational clinical studies or clinical trials (*Deb et al., 2016*). PSM method can effectively reduce the influence of bias and confounding variables in subgroup analysis of observational clinical studies and RCT studies, in order to make more reasonable comparisons between the observation group and the control group similar to randomized controlled trials (*Deb et al., 2016*). The advantage of PSM is to transform the control of confounding factors into the control of tendency value, so as to achieve the purpose of dimensionality reduction and control of confounding bias, and to solve the multi-dimensional problem of multiple non-study factors and the potential collinearity problem of independent variables (*Zakrison, Austin & Mccredie, 2018*). PSM is generally applicable to two types (*Deb et al., 2016*). First, in observational studies, the number of directly comparable individuals in the control group and the experimental group is small. Second, picking a subset from the control group that is same or similar to the experimental group in all parameters for comparison is very difficult because there are many parameters to measure individual characteristics. In the first type, the intersection between the experimental and control groups is small. For example, the healthiest (10%) in the treatment group is similar to the least healthy subgroup (10%) in the non-treatment group. When comparing these two overlapping subsets it leads to a very biased conclusion. PSM can be achieved by various statistical software, which is simple and convenient, does not increase the difficulty of matching, and can match multiple confounding factors at the same time (*Benedetto et al., 2018*; *Kuss, Blettner & Börgermann, 2016*). PSM method is increasingly being applied in the medical field as an effective and ideal statistical tool.

## METHODS

### Study design

We conducted a retrospective observational study involving patients with first-time AHF admissions at the Xiaoshan Hospital Affiliated to Hangzhou Normal University between January 2018 and December 2023. Based on whether or not receiving sacubitril/valsartan during hospitalization, two groups were set up: the treatment group and the control group. Patients in the treatment group received sacubitril/valsartan within 24 h of confirmed AHF

diagnosis. The inclusion criteria for participants were: a. age ≥ 18 years; b. AHF diagnosed according to *2021 ESC Guidelines for the Diagnosis and Treatment of Acute and Chronic Heart Failure* (*ESC Scientific Document Group, 2021*), with the cardiac function of grade *II* to *IV*. The exclusion criteria were: a. patients with severe trauma or history of major surgical operations on admission; b. patients with severe liver disease, infectious diseases, severe tuberculosis, or malignant tumors; c. patients who were not hospitalized for the first time; d. patients with incomplete clinical information. The clinical data of patients who met the inclusion criteria were collected, including age, gender, BMI, complications, etiological classification, main therapeutic drugs, and laboratory indicators which included brain natriuretic peptide (BNP), creatinine (Cr), alanine transaminase (ALT), aspartate transaminase (AST), high density lipoprotein (HDL), low density lipoprotein (LDL), triglyceride (TG), *etc*. The endpoint event of the study was whether the patient was readmitted within 6 months after discharge. The study protocol was approved by the Medical Ethics Committee of Xiaoshan Hospital Affiliated to Hangzhou Normal University (Approval number: K2023008). The study has obtained a waiver of the need for informed consent from participants.

## Statistical methods

The statistical analysis was completed in the SPSS 22.0 software package (IBM Corp., Armonk, NY, USA). The normally distributed measurement data were expressed as mean ± standard deviations (SD) and the independent samples *t*-test was used for inter group comparison. The normality was verified using Kolmogorov-Smirnov. The skewed distributed count data were represented by median and quartile, and the Mann-Whitney U test was used for inter group comparison. The count data were expressed by the number of cases and percentages, and the inter group comparison was conducted using the $\chi^2$ test. The PSM process was performed using the PSM extension program of the SPSS 22.0. With using sacubitril/valsartan as the dependent variable and each covariate as the independent variable, propensity score values were estimated through logistic regression. The covariates included age, sex, comorbidities (hypertension, diabetes, chronic obstructive pulmonary disease), smoking status, medications, body mass index (BMI), and laboratory parameters (BNP, ALT, AST, Cr, LDL, *etc*.). Patients with AHF of either ischemic or non-ischemic etiology were included in the study. A retrospective review of the patient's diagnostic and therapeutic course revealed that sacubitril/valsartan may be appropriately administered to patients with either ischemic or non-ischemic AHF. Thus, etiological classification was not incorporated as a covariate in the analysis. A 1:1 nearest neighbor matching method was performed for matching. This process ensured the quality of the matching results by using a caliper value of 0.2 (*R* language). Then the changes in the standard deviation of covariates between two groups before and after matching were compared. The closer the standard deviation after matching was to 0, the more satisfactory the matching result was. When the absolute value of the standard deviation was less than 0.1, it was considered that the balance of inter-group variables was good. Logistic regression analysis was conducted to investigate the effect of using sacubitril/valsartan on the readmission rate within 6 months in patients with AHF.

## RESULTS

### Baseline conditions before matching of two groups

A total of 176 patients with AHF were enrolled in our study, comprised of 143 (81.3%) with ischemic etiology and 33 (18.7%) with non-ischemic causes. Ischemic heart failure (IHF) is defined as myocardial dysfunction resulting from coronary artery disease (CAD)-induced ischemia or infarction, whereas non-ischemic heart failure (NIHF) encompasses etiologies independent of CAD, including cardiomyopathies, inflammatory/infectious processes, and metabolic disorders. In this study, both IHF and NIHF diagnoses were physician-adjudicated through comprehensive clinical evaluation. Among them, 86 cases received treatment with sacubitril/valsartan during the acute stage (the treatment group) and 90 cases did not receive sacubitril/valsartan (the control group). The baseline clinical characteristics of the cohorts prior to PSM were shown in Table 1. Compared with the control group, the proportion of β-receptor blockers used in the treatment group was significantly higher, while the proportion of diuretics and ACEIs was lower, with statistically significant differences.

### Baseline conditions after matching of two groups

Matching was performed using a 1:1 nearest neighbor approach with the caliper value set to 0.2 (R language). The treatment group was used as the baselined group for matching. A total of 53 pairs between the two groups were successfully matched. The imbalanced covariates between the two groups reached equilibrium after matching (Figs. 1, 2). The equilibrium of each covariate had been significantly improved through PSM method ($P > 0.05$), as shown in Table 2.

### Impact of sacubitril/valsartan on readmission within 6 months

Before matching, 32 of 176 patients (18.18%) were readmitted to hospital due to heart failure, including nine patients in the treatment group (10.47%) and 23 patients in the control group (25.56%). Using whether readmission as the dependent variable and whether to use sacubitril/valsartan as the independent variable, logistic regression analysis was performed. The result of univariate logistic regression analysis showed that the risk of readmission within 6 months in the treatment group was only 34% of that in the control group (OR = 0. 34, 95% CI [0.147–0.787], $P = 0.012$). After adjusting for age, sex, biochemical examination, clinical medication, *etc.*, the risk of readmission within 6 months in the treatment group was 33.9% of that in the control group (OR = 0.339, 95% CI [0.116–0.994], $P = 0.049$). These results indicated that the use of sacubitril/valsartan in AHF could reduce the risk of readmission, which was a protective factor. See Table 3 for details.

After PSM matching, 18 of 106 patients (53 patients in each group) were readmitted to hospital, including seven patients in the treatment group (13.21%) and 11 patients in the control group (20.75%). The readmission rate in the treatment group was lower than that in the control group, with no significant statistical difference between the two groups. Using whether to use sacubitril/valsartan as the independent variable and whether readmission as the dependent variable, univariate logistic regression and multivariate

**Table 1 Clinical features of patients in two groups before matching.** A total of 86 cases (48.86%) were treated with sacubitril/valsartan during the acute stage (the treatment group) and 90 cases were not received sacubitril/valsartan (the control group).

| General information | Total | Control group ($n$ = 90) | Treatment group ($n$ = 86) | Inspection method | $P$-value |
|---|---|---|---|---|---|
| Age (year) | 75.5 (64–84) | 78 (66–86) | 74 (60.5–82) | Whitney U | 0.074 |
| Male [$n$ (%)] | 91 (51.70) | 46 (51.11) | 45 (52.33) | $\chi^2$ test | 0.872 |
| Ischemic type [$n$ (%)] | 143 (81.25) | 71 (78.89) | 72 (83.72) | $\chi^2$ test | 0.412 |
| Hypertension [$n$ (%)] | 117 (66.48) | 55 (61.11) | 62 (72.09) | $\chi^2$ test | 0.123 |
| Diabetes [$n$ (%)] | 37 (21.02) | 16 (17.78) | 21 (24.42) | $\chi^2$ test | 0.280 |
| COPD [$n$ (%)] | 23 (13.07) | 10 (11.11) | 13 (15.12) | $\chi^2$ test | 0.431 |
| Smoking [$n$ (%)] | 65 (36.93) | 31 (34.44) | 34 (39.53) | $\chi^2$ test | 0.484 |
| Diuretic [$n$ (%)] | 170 (96.59) | 84 (93.33) | 86 (100.00) | Fisher | 0.029 |
| ACEI [$n$ (%)] | 25 (14.20) | 20 (22.22) | 5 (5.81) | $\chi^2$ test | 0.002 |
| ARB [$n$ (%)] | 46 (26.14) | 27 (30.00) | 19 (22.09) | $\chi^2$ test | 0.233 |
| CCB [$n$ (%)] | 47 (26.70) | 26 (28.89) | 21 (24.42) | $\chi^2$ test | 0.503 |
| β-receptor blocker [$n$ (%)] | 57 (32.39) | 23 (25.56) | 34 (39.53) | $\chi^2$ test | 0.048 |
| BMI (kg/m$^2$) | 22.89 (20.83–26.41) | 22.83 (20.26–24.26) | 23.22 (20.89–27.16) | Whitney U | 0.086 |
| BNP (pg/mL) | 891.6 (580.83–1530.1) | 854.8 (600.13–1411.1) | 947 (530.25–1638.75) | Whitney U | 0.657 |
| Cr (μmol/L) | 89.79 (71.38–112.8) | 91.35 (70.28–117.63) | 88.65 (72.1–111.13) | Whitney U | 0.775 |
| ALT (U/L) | 21.9 (16.1–36.7) | 20.65 (15.43–40.33) | 23 (16.25–34.15) | Whitney U | 0.438 |
| AST (U/L) | 27.35 (21–36.83) | 27.95 (21.83–36.73) | 26.7 (19.6–37.03) | Whitney U | 0.484 |
| TG (mmol/L) | 0.95 (0.69–1.27) | 0.95 (0.69–1.25) | 0.95 (0.69–1.27) | Whitney U | 0.77 |
| HDL (mmol/L) | 1.12 ± 0.3 | 1.12 ± 0.34 | 1.12 ± 0.25 | $t$-test | 0.921 |
| LDL (mmol/L) | 2.18 (1.68–2.74) | 2.09 (1.67–2.65) | 2.33 (1.69–2.97) | Whitney U | 0.154 |
| Cases of readmission within 6 months | 32 (18.18) | 23 (25.56) | 9 (10.47) | $\chi^2$ test | 0.009 |

**Note:**
COPD, chronic obstructive pulmonary disease; ACEI, angiotensin-converting enzyme inhibitor; ARB, Angiotensin receptor antagonist; CCB, Calcium channel blocker.

logistic regression were performed. The results of univariate logistic regression (OR = 0.699, 95% CI [0.267–1.831], $P$ = 0.466) and multivariate logistic regression (OR = 0.728, 95% CI [0.212–2.496], $P$ = 0.252) showed no statistical significance, suggesting that the use of sacubitril/valsartan in AHF did not reduce the risk of readmission and was not a protective factor. The results were shown in Table 4.

## DISCUSSION

As an observational study, it is obvious that random grouping cannot be achieved at the initial stage, and confounding factors inevitably existed in the study, which affected the accuracy of the results. Therefore, the PSM method was involved in this study to reduce the bias brought by samples. However, although the PSM method can control the confounding factors between groups to make the balance comparable, it will also lead to sample size reduction (*Johnson et al., 2018*). A total of 176 patients were initially included in this study, and the readmission rate within 6 months was 18.18%, which was consistent with the results of previous studies (*Chioncel et al., 2017*). After PSM matching, the sample size was obviously reduced. A total of 53 matched pairs ($n$ = 106) were successfully generated

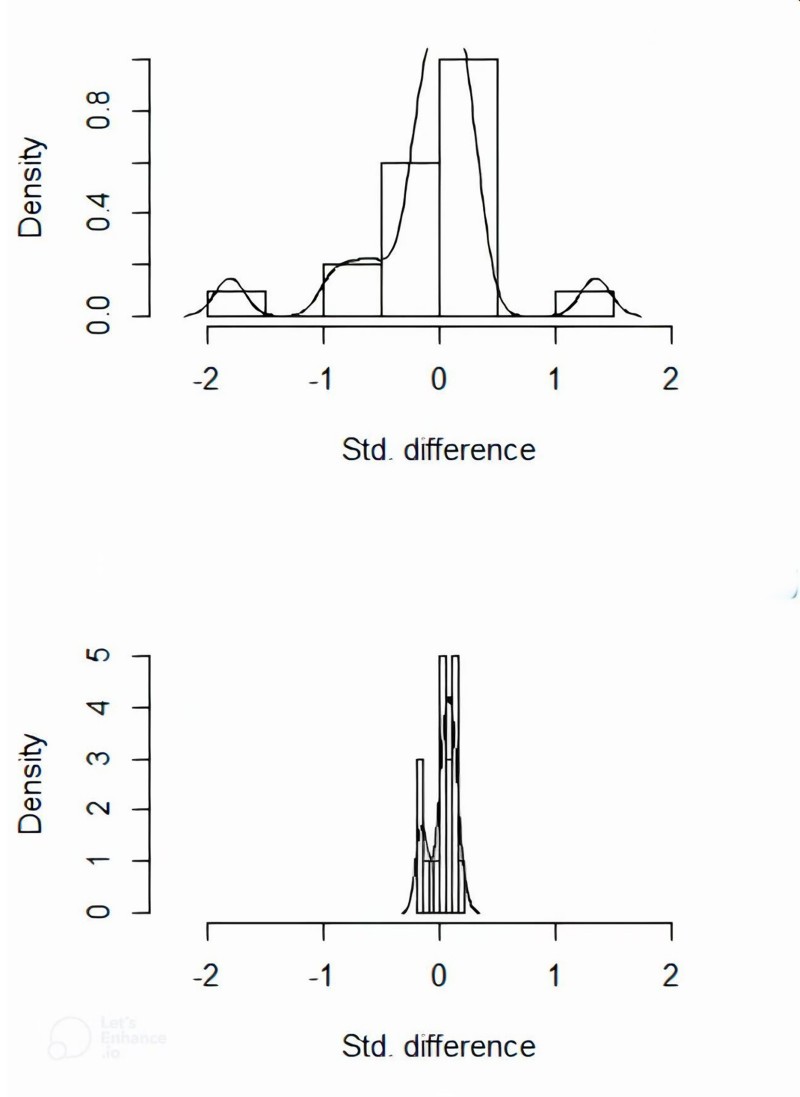

**Figure 1 Distribution histogram of the standard deviations before and after matching.** The imbalanced covariates between the two groups reached equilibrium after matching.

through PSM. The reduced sample size following matching may limit the statistical power and generalizability of our findings, potentially increasing the risk of Type II errors. Among the 176 enrolled patients, all were clinically evaluated by physicians as eligible for sacubitril/valsartan therapy with no documented contraindications. Treatment allocation was ultimately determined by non-medical factors in control group participants. In the initial phase of the study, our hospital had not yet formally introduced sacubitril/valsartan. During the latter half of the study, after sacubitril/valsartan was officially made available in the hospital, patients with AHF without contraindications were treated with this medication.

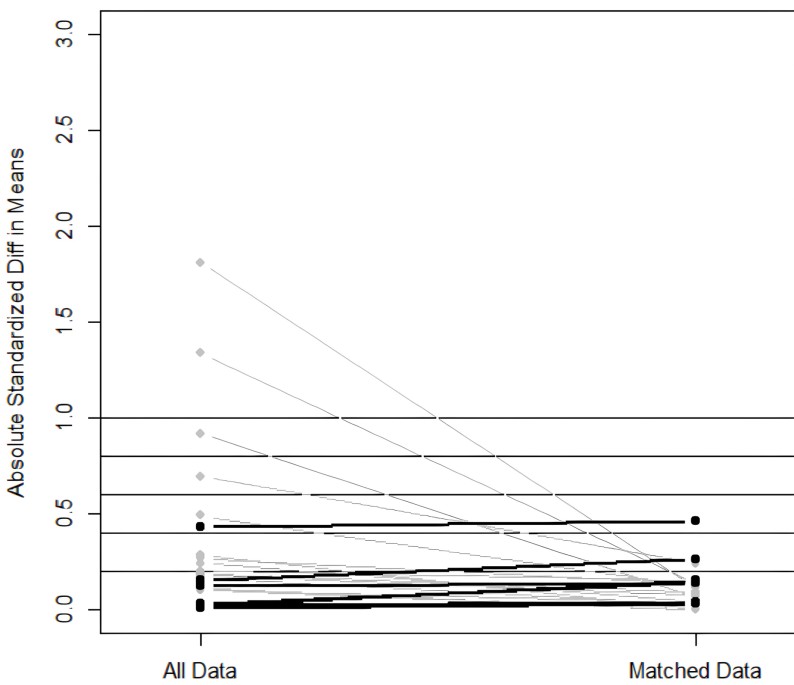

**Figure 2 Scatter plot of standardized mean differences before and after matching.** The imbalanced covariates between the two groups reached equilibrium after matching.

**Table 2 Clinical features of patients in two groups after matching.** A total of 53 pairs between the two groups were successfully matched.

| General information | Total | Control group ($n = 53$) | Treatment group ($n = 53$) | Inspection method | P-value |
|---|---|---|---|---|---|
| Age (year) | 74.5 (64–84.25) | 74 (66.5–85.5) | 75 (62.5–84) | Whitney U | 0.857 |
| Male [$n$ (%)] | 57 (53.77) | 28 (52.83) | 29 (54.72) | $\chi^2$ test | 0.846 |
| Ischemic type [$n$ (%)] | 86 (81.13) | 41 (77.36) | 45 (84.91) | $\chi^2$ test | 0.321 |
| Hypertension [$n$ (%)] | 73 (68.87) | 37 (69.81) | 36 (67.92) | $\chi^2$ test | 0.834 |
| Diabetes [$n$ (%)] | 23 (21.70) | 12 (22.64) | 11 (20.75) | $\chi^2$ test | 0.814 |
| COPD [$n$ (%)] | 14 (13.21) | 7 (13.21) | 7 (13.21) | $\chi^2$ test | 1 |
| Smoking [$n$ (%)] | 42 (39.62) | 20 (37.74) | 22 (41.51) | $\chi^2$ test | 0.691 |
| Diuretic [$n$ (%)] | 106 (100) | 53 (100) | 53 (100) | – | – |
| ACEI [$n$ (%)] | 12 (11.32) | 7 (13.21) | 5 (9.43) | $\chi^2$ test | 0.54 |
| ARB [$n$ (%)] | 30 (28.30) | 17 (32.08) | 13 (24.53) | $\chi^2$ test | 0.388 |
| CCB [$n$ (%)] | 25 (23.58) | 14 (26.42) | 11 (20.75) | $\chi^2$ test | 0.492 |
| β-receptor blocker [$n$ (%)] | 36 (33.96) | 20 (37.74) | 16 (30.19) | $\chi^2$ test | 0.412 |
| BMI (kg/m$^2$) | 23.16 (20.99–26.65) | 23.18 (22.04–26.62) | 22.22 (20.56–26.66) | Whitney U | 0.194 |
| BNP (pg/mL) | 880.1 (528.15–1411.1) | 821.4 (546.3–1235.1) | 998.8 (519.2–1588.05) | Whitney U | 0.229 |
| Cr (μmol/L) | 84.15 (68.75–110.95) | 82.9 (68.65–111.85) | 89.5 (74.8–113.85) | Whitney U | 0.153 |
| ALT (U/L) | 20.45 (15.78–29.45) | 20 (15.35–29) | 20.8 (16.2–30.2) | Whitney U | 0.645 |
| AST (U/L) | 26.15 (20.6–34.9) | 24.8 (19.45–33.6) | 26.8 (20.95–36.6) | Whitney U | 0.281 |

(Continued)

| Table 2 (continued) | | | | | |
|---|---|---|---|---|---|
| General information | Total | Control group (*n* = 53) | Treatment group (*n* = 53) | Inspection method | *P*-value |
| TG (mmol/L) | 0.94 (0.68–1.22) | 0.93 (0.72–1.21) | 0.94 (0.68–1.26) | Whitney U | 0.783 |
| HDL (mmol/L) | 0.14 ± 0.30 | 1.14 ± 0.34 | 1.14 ± 0.26 | *t*- test | 0.897 |
| LDL (mmol/L) | 2.13 (1.66–2.83) | 2.12 (1.64–2.74) | 2.15 (1.64–2.97) | Whitney U | 0.714 |
| Cases of readmission within 6 months | 18 (16.98) | 11 (20.75) | 7 (13.21) | $\chi^2$ test | 0.301 |

Note:
COPD, chronic obstructive pulmonary disease; ACEI, angiotensin-converting enzyme inhibitor; ARB, Angiotensin receptor antagonist; CCB, Calcium channel blocker.

**Table 3 Results of univariate and multivariate logistic regression analysis before matching.** The result of univariate logistic regression analysis showed that the risk of readmission within 6 months in the treatment group was only 34% of that in the control group (OR = 0. 34, 95% CI [0.147–0.787], *P* = 0.012).

| Model | OR | *P*-value |
|---|---|---|
| Univariate logistic regression | 0.34 (0.147–0.787) | 0.012 |
| Multivariate logistic regression | 0.339 (0.116–0.994) | 0.049 |

**Table 4 Results of univariate and multivariate logistic regression analysis after PSM matching.** The readmission rate in the treatment group was lower than that in the control group, with no significant statistical difference between the two groups.

| Model | OR | *P*-value |
|---|---|---|
| Univariate logistic regression | 0.699 (0.267–1.831) | 0.466 |
| Multivariate logistic regression | 0.728 (0.212–2.496) | 0.252 |

While the crude analysis suggested a potential benefit of sacubitril/valsartan on 6-month readmissions, the propensity-matched comparison failed to confirm this effect, indicating possible residual confounding in the initial observational data. Prior to matching, both univariate and multivariate binary logistic regression showed that sacubitril/valsartan was a protective factor for reducing readmission to the hospital after initial diagnosis of AHF. After matching, neither univariate logistic regression analysis nor multivariate logistic regression analysis showed that sacubitril/valsartan used during hospitalization was a protective factor for reducing readmission of patients. Although the treatment group exhibited a lower readmission rate, no statistically significant difference was observed between the two groups for the endpoint event.

Presently, there are few studies on the use of sacubitril/valsartan for AHF. In the PIONEER-HF study (*McMurray et al., 2014*; *Morrow et al., 2019*; *Berg et al., 2021*; *Velazquez et al., 2018*; *Devore et al., 2020*), heart failure with reduced ejection fraction (HFrEF) patients hospitalized for new heart failure or acute decompensated heart failure (ADHF) were randomly given sacubitril/valsartan or enalapril after stabilization. A greater reduction in N-terminal pro-B-type natriuretic peptide (NT-proBNP) was observed at 4 and 8 weeks in the treatment group, with fewer adverse events related to heart failure, providing evidence for the effectiveness of in-hospital use of sacubitril/valsartan. In the

open-label TRANSITION study (*Pascual-Figal et al., 2018*; *Wachter et al., 2019*), more than 1,000 HFrEF patients hospitalized with ADHF were randomly assigned to early in-hospital medication group (sacubitril/valsartan used after 24 h of hemodynamic stabilization) or post-discharge medication group (sacubitril/valsartan used within 14 days of discharge). The use of sacubitril/valsartan at different stages showed the same security. Another study in newly diagnosed patients with AHF suggested that early intervention with sacubitril/valsartan is thought to delay disease progression in patients with neonatal HFrEF (*Senni et al., 2020*).

However, the results of our study are inconsistent with the above studies. We speculate that there are two possible reasons. First, the primary endpoint of our study is the readmission rate. Currently, clinical studies related to sacubitril/valsartan have not yet used readmission rates as an indicator of drug efficacy. Second, this study is a retrospective observational study in the real world without any intervention. PMS allows for the comparison of patients under relatively equitable conditions, providing more reference value than general regression studies. Therefore, we can assume that the matched results are more reliable. The use of sacubitril/valsartan in AHF may not be a protective factor for reducing readmission in patients. Notably, ejection fraction data were largely missing for the majority of patients. This unavoidable limitation may affect the generalizability of our findings.

## CONCLUSIONS

In conclusion, the results of this study suggest that the use of sacubitril/valsartan during hospitalization in newly diagnosed AHF does not reduce the risk of readmissions. However, there are certain limitations in the study, more clinical studies are needed to support this conclusion. As a retrospective study, some clinical indicators such as BNP, ejection fraction, *etc*., are missing, leading to no comparison of the indicators before and after 6 months. In addition, the study is a single-center study with a small sample size. The admission status of patients in other medical institutions cannot be obtained. The follow-up multi-center observation studies should be conducted for further discussion. For similar observational studies, PSM method can greatly reduce the bias of retrospectively collected data, thus obtaining more reliable results.

## ACKNOWLEDGEMENTS

The authors used letsenhance.io to enlarge Fig. 1; no AI was used to generate the data or figure.

### Funding

This work was supported by the Hangzhou Pharmaceutical Association Hospital Pharmaceutical Innovation Research Fund Project (Grant/Award Number: N/A). The funders had no role in study design, data collection and analysis, decision to publish, or preparation of the manuscript.

## Grant Disclosures

The following grant information was disclosed by the authors:
Hangzhou Pharmaceutical Association Hospital Pharmaceutical Innovation Research Fund.

## Competing Interests

The authors declare that they have no competing interests.

## Author Contributions

- Lingdi Zhang performed the experiments, prepared figures and/or tables, and approved the final draft.
- Qinglian Chen analyzed the data, authored or reviewed drafts of the article, and approved the final draft.
- Guofei Ren conceived and designed the experiments, authored or reviewed drafts of the article, and approved the final draft.

## Human Ethics

The following information was supplied relating to ethical approvals (*i.e.*, approving body and any reference numbers):

The Medical Ethics Committee of Xiaoshan Hospital Affiliated to Hangzhou Normal University (Approval number: K2023008).

## Data Availability

The raw measurements are available in the Supplemental File.

## Supplemental Information

Supplemental information for this article can be found online at http://dx.doi.org/10.7717/peerj.20055#supplemental-information.

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
