# Peer review of "A propensity score-matched retrospective study on sacubitril/valsartan use and 6-month readmission rates in patients with acute heart failure"

_PeerJ, doi:10.7717/peerj.20055_

## Round 0.1 · original submission · Major Revisions

All 3 reviews are brief, but in sum they make valuable points. Please respond to their comments

Reviewer 1 ·

Basic reporting

The article is well written. The literature reference is acceptable.
The table when the medication was initiated.
Only if these are analysed as different variables, the results could be analysed in the clinical context

Experimental design

The research question is well defined. However the important information on what categories of heart failure it was studied was not given. This is because of non availabilty of ejection fraction by echo. The data available on ARNI in HFrEF and HFpEF are very different.Until and unless this data is available ,the study may not look meaningful. One suggestion would be only to analyse the data on HFrEF ,based on available LVEF data.
This is a serious limitation.

Validity of the findings

The validity of the findings should have been interpreted in the context of various categories of heart failure. The multivariate analysis should have included the clinical parameters mentioned earlier.

Additional comments

No further comments.

·

Basic reporting

The article is written in clear, technically accurate English and adheres to professional standards. It includes an introduction and background to show its relevance in the broader field of knowledge and reference relevant prior literature appropriately.

Comment: Remove "Entresto" and use either Sacubitril/Valsartan" or "Angiotensin Receptor-Neprilysin Inhibitors"

Experimental design

The propensity matching should include the age, sex, etiology of HF, and co-morbidities (Diabetes, COPD, Obesity, etc)

Validity of the findings

This is a retrospective study. Please compare the outcomes to prior Clinical trials where ARNI has been shown to reduce hospitalization and explain the negative impact of the drug in this study and why after propensity matching, there was no reduction in hospitalization rate.

Additional comments

Explain how the subjects are chosen to receive ARNI?

Reviewer 3 ·

Basic reporting

No comment

Experimental design

No comment

Validity of the findings

Authors mentioed that---Therefore, we can assume that the matched results are more reliable (Line: 170). What is the basis and reference used to validate this statement.
How we can generalize the results in a population keeping similar RCTs in mind?

---

## Round 0.2 · Major Revisions

Please see the reviewer suggestions under "Additional comments".

·

Basic reporting

Strengths:
• The manuscript is generally written in clear and professional English.
• The introduction and background provide adequate context and include relevant citations from key studies such as PIONEER-HF and TRANSITION.
• The data is supported by raw tables and figures, and results are described with appropriate statistical references.
• The manuscript is structured logically and adheres to scientific reporting norms.

Experimental design

Strengths:
• Clearly defined study design: retrospective observational with appropriate use of propensity score matching (PSM).
• Ethical approval is obtained and documented.
• Well-defined inclusion and exclusion criteria.
• Adequate justification is provided for the use of PSM to minimize bias in observational settings.

Validity of the findings

Strengths:
• Results are clearly presented before and after PSM.
• Both univariate and multivariate analyses were conducted appropriately.
• The authors acknowledge the limitations of real-world data and sample size reduction due to PSM.

Additional comments

Major Suggestions:
1. Improve clarity on covariates used in PSM.
2. Clarify timing of intervention (when during hospitalization the drug was started).
3. Revise figure/visual presentation for improved readability.
4. Consider a sentence to clarify the “non-significant trend toward benefit” in post-matching results — readers might infer clinical but not statistical benefit.

---

## Round 0.3 · Minor Revisions

Dear Authors,

Thank you for submitting your manuscript to PeerJ. After peer review, we are pleased to inform you that your submission has been assessed as requiring minor revisions prior to further consideration for publication.

The reviewer had no specific concerns regarding the basic reporting or the validity of the findings, which is encouraging. However, they highlighted a few issues related to the experimental design that should be addressed in your revised manuscript:

1. The distinction between ischemic and non-ischemic heart failure in your patient population is not clearly stated and should be clarified in the Methods and Results sections.

2. The lack of echocardiographic parameters in the methodology was noted. While this does not invalidate your findings, it is important to explicitly acknowledge this omission and its potential impact.

3. The sample size was considered small for a robust propensity score matching (PSM) analysis. Please justify your approach or discuss the limitations associated with it.

Additionally, we kindly ask that you include a more explicit limitations section addressing the points above. The reviewer also suggested that your discussion would benefit from reporting the baseline proportion of patients eligible for ARNI therapy, as well as exploring the reasons why some patients did not receive ARNI during the study period, which may offer valuable insights for interpreting your comparisons.

We look forward to receiving your revised manuscript, along with a detailed point-by-point response to the reviewer’s comments.

Reviewer 1 ·

Basic reporting

No comment

Experimental design

I had mentioned that the methodology does not involve echo parameters. Moreover, the ischemic/non-ischemic HF is not clear. The numbers are very small for a PSM analysis.

Validity of the findings

No comment

Additional comments

Could you include limitations based on the comments given above?

---

## Round 0.4 · Minor Revisions

Thanks for submitting your revised manuscript, there are a couple of small matters that still need attention.

Firstly, one reviewer asked about how ischaemic and non-ischaemic heart failure were distinguished in your population. I see that you have recorded the proportion of patients with ischaemic heart failure, but you have not described how you made the distinction. Please do this.

Secondly, the comments of the reviewers about not having data about ejection fraction really just need a statement to the effect that having this data would have strengthened the conclusions of the study, but that unfortunately it was not available. I don't think that the several reasons as to why you didn't have the data are important, or add to the manuscript, and I think they should be removed.

Reviewer 1 ·

Basic reporting

-

Experimental design

Lack of an echocardiogram is a significant drawback.

Validity of the findings

-

---

## Round 0.5 · accepted · Accept

Thank you for making these final minor revisions.